# International Guidelines for Hypertension: Resemblance, Divergence and Inconsistencies

**DOI:** 10.3390/jcm11071975

**Published:** 2022-04-01

**Authors:** Junior Justin, Antoine Fayol, Rosa-Maria Bruno, Hakim Khettab, Pierre Boutouyrie

**Affiliations:** 1Pharmacologie HEGP, APHP (Assistance Publique Hôpitaux de Paris), 75015 Paris, France; junior.justin@aphp.fr (J.J.); antoine.fayol@aphp.fr (A.F.); rosa-maria.bruno@inserm.fr (R.-M.B.); hakim.khettab@aphp.fr (H.K.); 2PARCC (Paris Cardiovascular Reseach Center), INSERM (Institut National de la Santé et Recherche Médicale) U970 Team 7, 75015 Paris, France; 3Department of Pharmacology, Université de Paris, 75015 Paris, France

**Keywords:** hypertension, antihypertensive agents, aging, cardiovascular diseases, vascular stiffness, guidelines

## Abstract

High blood pressure is the number one killer in the world. About 1.5 billion people suffered from hypertension in 2010, and these numbers are increasing year by year. The basics of the management of high blood pressure are described in the Canadian, American, International and European guidelines for hypertension. However, there are similarities and differences in the definition, measurement and management of blood pressure between these different guidelines. According to the Canadian guidelines, normal blood pressure is less than 140/90 mmHg (systolic blood pressure/diastolic blood pressure). The AHA and ESC estimate normal blood pressure to be less than 120/80 mmHg (systolic blood pressure/diastolic blood pressure). Regarding treatments, the AHA, ISH and ESC are also in agreement about dual therapy as the first-line therapy, while Canadian recommendations retain the idea of monotherapy as the initiation of treatment. When it comes to measuring blood pressure, the four entities agree on the stratification of intervention in absolute cardiovascular risk.

## 1. Introduction

High blood pressure (HBP) is the leading preventable cause of cardiovascular death in the world. Stroke is the second cause of death and third cause of disability in the world; however, HBP is the primary risk factor for stroke [1]. In 2010, it was estimated that 1.4 billion people had high blood pressure, which represents 31% of the world’s adult population [2]. The trend is increasing, particularly as a result of emerging countries and the aging of populations; some international bodies expect an increase of 10 to 15% by 2025 [3]. Hypertension, therefore, remains a global public health problem.

Four international recommendations on high blood pressure were published in 2017, 2018 and 2020, and form the basis for hypertensive patient management: the Canadian and American recommendations published in 2017 [4,5], the European recommendations published in 2018 [3] and the International Society of Hypertension in 2020 [6]. The questions of the relevance within guidelines and discrepancies between them have been extensively commented upon [7,8,9]. It is not our aim to replicate those detailed analyses, but rather, to expose some of the most relevant concordances and divergence between the four guidelines, and to determine which of them may lead to inconsistencies.

## 2. Diagnosis of Hypertension

While all four societies agree that high blood pressure is a major public health problem, it is not defined in the same way by each of them. Table 1 summarizes the different definitions used, as well as the treatment objectives according to the various recent international recommendations.

The definition and objectives of treatment are different according to international recommendations. Lowering the definition of high blood pressure in the United States mechanically increased the number of hypertensive patients from 32% of the adult population to 46% [4]. The AHA/ACC justifies the threshold of 130/80 by the fact that the risk of cardiovascular disease increases exponentially from a systolic pressure of 115 mmHg up to >180 mmHg. An optimal control of the blood pressure figures of patients would theoretically reduce the number of complications related to arterial hypertension, especially since the prevalence of moderately elevated blood pressure is very high. This lowering of the blood pressure threshold is justified by the results of the SPRINT study [10], which compared two blood pressure targets in hypertensive patients: an intensive control with a blood pressure target <120/80 mmHg and a conventional control with a blood pressure target <140/90 mmHg. The SPRINT study demonstrated that treating hypertensive patients with a blood pressure target <120/80 mmHg decreases the risk of cardiovascular events as well as mortality, but at the expense of a significant increase in symptomatic hypotension, syncope and worsening ofkidney function [10]. One additional reason for decreasing the threshold might be related to the poor health status of the US population, with high incidence of sedentariness, obesity, diabetes, and social precarity leading to an overall reduction in life expectancy over the last 10 years, especially because of hypertension-related diseases [11]. The ESC, for its part, still considers BP values between 130/80 and 140/90 as pre-hypertensive, and does not recommend the introduction of pharmacological treatment for those patients considered to be hypertensive grade I according to the AHA; however, it recommends more regular control of blood pressure and the application of lifestyle and dietetic rules. Canadians do not make distinctions among normal BP values, with an unchanged threshold of 140/90 mmHg. The ISH is globally similar to the ESH; however, they do not define optimal BP.

Management also differs for diabetic patients. All guidelines agree that diabetes is a strong risk modifier, and that there is little evidence of the optimal level of BP being attained. Although high BP aggravates chronic kidney disease progression and represents an important target for treatment, at the time of writing of the guidelines, strict BP control was more harmful than protective, mainly based on the ACCORD trial [12]. Accordingly, the target BP in T2D patients is higher than in the rest of the population. The fact that more recent meta-analyses [13,14] call for stricter BP control in T2D patients is interesting.

An essential point concerns the measurement of BP. The measurement taken in the doctor’s office remains the reference for the four recommendations, against all scientific logic, but with pragmatism. Indeed, office BP measurement is universal, and most of the scientific evidence relies on such measures, despite the massive flaws related to white-coat hypertension, masked hypertension, a lack of standardization, and the demonstrated superiority of other modalities (unattended, home or ambulatory) of measurements. No guideline elaborates on the limits of BP monitors, despite alarming reports [15], with the motto of scientific societies being that it is better to measure with bad devices than to not measure at all. The results of SPRINT, greatly influencing the AHA/ACC recommendations, were obtained with an automatic measurement in the absence of caregivers (“unattended”), which provides typical readings lower than those in the office. This approach has been widely criticized for being poorly referenced and little-used in clinical practice. Measures outside the office (self-measurement, outpatient measure, pharmacy, etc.) are discussed by the four recommendations but remain marginalized, even though they are the most relevant, particularly for the initiation of treatment. Pragmatism dictates that bad measurements are better than no measurement at all. The four learned societies hesitate to take a clear position on this subject, which is regrettable.

The four international recommendations converge on the need to integrate blood pressure into an overall assessment of the patient’s cardiovascular risk, which should help prioritize the timing of pharmacological intervention and its intensity, in line with the position of the Lancet Commission on Hypertension [16]. Intervention is preconized earlier (and more intensely) in the case of secondary prevention (overt CVD), diabetes, or highly integrated CV risk. Indeed, high blood pressure is, in fact, a proxy for a more generalized cardiovascular disease involving hypertension-mediated organ damage (HMOD)—such as direct damage to the large and small arteries [17], heart, brain and kidneys [18]—which can be summarized as early vascular aging [19]. Although earlier and more intense treatment of patients with overt CVD or diabetes, or who are at very high risk, it would have been logical to propose earlier identification of HMOD for guiding treatment, rather than waiting for overt CVD. This can point to a lack of political courage by committees. Indeed, the evaluation of the damage to the target organs (heart, kidney, arteries, and brain) is used for risk evaluation (and thus, for the initiation of treatment), and the use of HMOD assessment was degraded to 2B evidence (i.e., “may be considered”). This lack of logic was highlighted during the drafting of the ESH/ESC recommendations, but has not been corrected for.

The management of a hypertensive patient is based on two main axes: non-pharmacological measures and pharmacological measures. The four international recommendations converge on the non-pharmacological management of the patient. The main recommendations are listed below:Weight reduction for overweight patients;Adaptation of the diet;Reduction in salt consumption;Practice of regular physical activity;Reduction in alcohol consumption;Smoking cessation.

All of these measures should be recalled at each consultation. On average, the correction of each of these factors causes a drop of 4 to 5 mmHg in systolic pressure. The rules for the implementation of these recommendations are elusive; in particular, there is no incentive that these rules should be prescribed (as a full-fledged treatment). Although symbolic, and with low evidence for efficacy, this recommendation would have been very helpful.

For pharmacological management, all of the therapeutic strategies have recently been modified. The main novelty is the first-line introduction of dual therapy in the recommendations of the ESH, ISH and AHA/ACC. This is justified because the combination of two drug classes is about five times more effective in lowering blood pressure levels, compared to doubling the dose of a monotherapy [20]. This allows for faster and more consistent blood pressure control in the patient. Finally, the use of therapeutic combinations of two complementary pharmacological classes makes it possible to use lower doses and can reduce the dose-dependent adverse effects of each class, thus improving the acceptability of the treatments. The industrial development of fixed combinations makes it possible to limit the number of pills prescribed, and may promote adherence. The Canadian recommendations are more conservative and still recommend monotherapy at initiation. The use of dual therapy is one among other possibilities offered by the ISH. One can wonder why. We have no clear explanation for Canada. The ISH guidelines have a special focus on low–middle income countries, where the cost of treatment might be an explanation.

Figure 1 transcribes the first three stages of treatment according to the various international recommendations.

The choice of drug classes is left free and is based on the three main classes (RAS blockers, calcium channel blockers, and diuretics), with indications based on the elective properties of each of the classes and the possible presence of comorbidities or the notion of prior tolerance. Angiotensin-converting enzyme inhibitors (ACEi) and angiotensin 2 receptor antagonists (ARB) have practically identical effectiveness, whether on BP control, total mortality or on the occurrence of cardiovascular events. Both classes are very effective at delaying chronic kidney disease, notably in diabetics, as underlined by the four guidelines. Regarding clinical tolerance, ARBs seem to be better tolerated by patients than ACEis, and lead to fewer discontinuations of treatment due to adverse effects [21]; however, the distinction between ACEis and ARBs is hotly discussed, not always on scientific grounds. Another important change concerns the downgrade of beta blockers. In the previous recommendations, beta-blockers were put at the same level as other available therapies [22]. Currently, the ESC and the AHA recommend their use as a fourth line after the introduction of anti-aldosterone, outside elective indications (angina, heart failure, and arrhythmias), because they were regularly shown to be less effective than other classes. Again, all four guidelines do not enter into the complexity of the pharmacology of beta-blockers, which is a pity, since the relative lack of efficacy of beta-blockers could be restricted to non-selective or beta1-selective antagonists, and do not concern vasodilating beta-blockers.

## 3. What to Do: What Do We Tell the Patient?

The different international guidelines use slightly different definitions of hypertension, but seem to agree on the therapeutic strategies to be used in the first line. However, they generate many difficulties for the patient. When should you be diagnosed with hypertension? The AHA/ACC recommendations, by lowering the threshold defining arterial hypertension, lead to a considerable increase in the number of affected patients and can put health systems in jeopardy. This might be acceptable in the US, where most of the financial effort is deferred to patients and private insurers, at the risk of further increasing social inequalities [23]. If it is to be applied in more regulated health-care systems, it would need to be further investigated. Mechanically increasing the number of hypertensive patients without having a strong policy aimed at reducing all of the cardiovascular risk factors (primordial prevention) will not lead to improved care to patients. It is certainly necessary to upgrade lifestyle and dietetic rules (prescribing them on prescription, better remuneration for preventive consultations, and eventually promoting healthcare pathways for wide cardiovascular prevention), so as to provide effective communication to patients newly identified as hypertensive.

The four guidelines are very conservative concerning the measurement of BP by not downgrading office BP measurement. This is a major problem, since BP measurement is now increasingly carried out outside the medical office, and out-of-office measurement reclassifies the patient (proven hypertension, white-coat, or masked hypertension), with major consequences for management. The multiplication of measures without appropriate pedagogy also leads to patient confusion, since they are not in a position to understand the figures delivered to them. The subject is addressed by the recommendations, but is not accompanied by a clear position, which is really a missed opportunity.

The therapeutic objectives are also limited to the control of BP, and therefore come up against the methodological difficulties raised when discussing diagnosis. There is a lack of objective criteria for initiating treatment, regulating its uptitration, and judging its effectiveness. The challenge in the coming years will be to find substitution criteria that make it possible to decide when to initiate pharmacological treatment, by combining optimal measurement of BP with HMOD using those new criteria for the uptitration (or reduction) of antihypertensive treatment. HMOD such as left-ventricular hypertrophy, microalbuminuria, and arterial stiffness may be used for this purpose. For the sake of illustration, pulse wave velocity (PWV) could change the way in which we manage our patients. Indeed, PWV reflects all of the cardiovascular risk factors, among them high blood pressure [24]; it can be measured non-invasively, is reproducible, and is less influenced by spurious changes than BP [25,26]. Having an objective measurement makes it possible to construct a therapeutic program with the patient in order to normalize the PWV, to compensate for diagnostic errors (white-coat and masked hypertension), and to detect poor blood pressure control in hypertensive patients. Does normalizing PWV improve blood pressure control and decrease the number of serious cardiovascular events? The recently published French multicenter SPARTE study [27] provides arguments in favor of such management, because patients treated by targeting PWV (<10 m/s) had better BP control and reduction in early vascular aging, compared to those treated by targeting blood pressure [28]. Finally, a major source of inconsistency within, and discrepancies between, guidelines can be found in the methodology of writing, especially because of conflicts of interest (COIs) affecting contributors [29]. COIs can be of different natures—industrial (often disclosed), economic, academic and institutional (almost never disclosed)—but in any case, they tend to bias the guidelines, at least in Japan [30], and likely in all situations. Unfortunately, disclosing does not appear to be a solution to limiting bias [29]. Although imperfect, the present methodology is next to best.

## 4. Conclusions

High blood pressure affects more than 1.5 billion people worldwide and the number is on the rise, especially in LMIC, and is largely due to shifting of the definition of hypertension to lower values. International guidelines use different thresholds to define the hypertensive patient, and all four of them minimize the methodological issues related to BP measurement. The main novelty of the international recommendations is the introduction of fixed-dose association therapies using pharmacologically synergistic drugs during the initiation of treatment, at least for the AHA/ACC and ESH. The semantic shift from hypertension taken in isolation to hypertension fitting into the larger picture of cardiovascular prevention is underway [16]; however, it is insufficiently taken up by the recommendations, which remain very conservative.

## Figures and Tables

**Figure 1 jcm-11-01975-f001:**
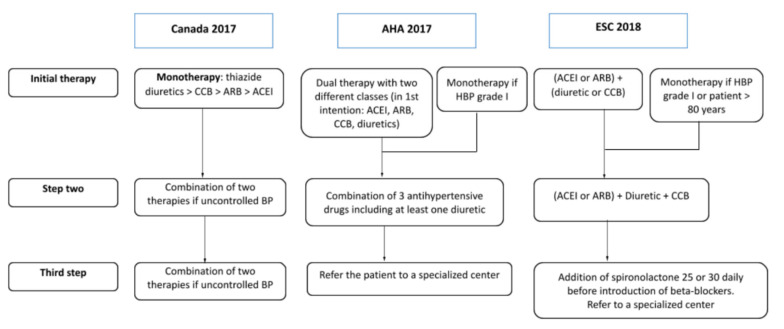
Therapeutic schedule for a patient with essential hypertension without other risk factors. * No ACEI/ARB in women with or planning pregnancy.

**Table 1 jcm-11-01975-t001:** Summary of hypertension grades and treatment objectives for a patient with essential high blood pressure.

Recommendations	Canada 2017				AHA 2017				ESC 2018				ISH 2020			
Categories		Systolic		diastolic		Systolic		diastolic		Systolic		diastolic		Systolic		diastolic
		(mmHg)		(mmHg)		(mmHg)		(mmHg)		(mmHg)		(mmHg)		(mmHg)		(mmHg)
	Normal BP	<140	and	<90	Normal BP	<120		<80	Optimal BP	<120	and	<80	Normal BP	<130		<85
	Grade I HBP	140–159	and/or	90–99	Elevated BP	120–129		<80	Normal BP	120–129	and/or	80–84	High-normal BP	130–139		85–89
	Grade II HBP	≥160	and/or	>100	Grade I HBP	130–139	or	80–89	Normal high	130–139	and/or	85–89	Hypertension BP	≥140		≥90
	Diabetic BP level: inconclusive evidence for diabetics for strict control				Grade II HBP	≥140	or	≥90	Grade I HBP	140–159	and/or	90–99	Grade I hypertension	140–159		90–99
					Diabetic BP level: inclusive evidence for strict control				Grade II HBP	160–179	and/or	100–109	Grade II hypertension	≥160		≥100
									Grade III HBP	≥180	and/or	≥110	Diabetic BP level:	<130		<80
									Diabetic BP level: <140, targeting 130, no evidence for lower							
CV risk assessment	Assessment of the overall CV risk of the patient is recommended.				Assessment of the overall CV risk of the patient is recommended.				CV risk assessment with risk score by SCORE is recommended (grade IB)				Assessment of the overall CV risk of the patient is recommended.			
Purpose of treatment	Recommended BP: <140/90 mmHg				Recommended BP: <130/80 mmHg				Recommended BP: <140/90 mmHg and if well tolerated BP <130/80 mmHg				Recommended BP: <130/80 mmHg if well tolerated (<65 years) and <140/90 mmHg in people (65 years if well tolerated.			

## Data Availability

Not applicable.

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
