# Peer review of "International Guidelines for Hypertension: Resemblance, Divergence and Inconsistencies"

_jcm, 2022, doi:10.3390/jcm11071975_

Round 1

Reviewer 1 Report

This study is important and documents important and confusing differences between different hypertension guidelines.  However, it is not the first to do this and as a minimum should document all published articles identifying this problem.  

The problem with Hypertension Guidelines is that they are produced by authors and organizations that are strongly conflicted by drug companies, device manufacturers and clinicians who stand to gain from the recommendations.  It is this bias that needs to be documented and may well be the main reason for the major differences in the recommendations.  

I found it hard to read and did not understand the rationale for many of the statements.  

Author Response

We thank the reviewer for his(her) comments. We do not claim to be the first to address the question of discrepancies between guidelines, but we did what we were asked to do, i.e. a summary of the main differences and their implications. We have nevertheless added 2 references in major journals addressing the question, and added the following sentence page xx, line yy "The question of the rationale for guidelines and discrepances have been extensively commented ref 1 ref 2). (10.1016/j.ejim.2019.01.016, doi: 10.1007/s40292-017-0236-x.

As for the adequation of guidelines and the role of conflict of interests, much can be said, not only on industrial conflict of interest. Many authors contributing to guidelines have also academic and institutional conflicts of interest which bias their views as much as industrial interest. For instance, considerations on practicality and global health economics take often advantage over scientific evidence. Just to illustrate this point, consider the marginal place of ABPM for diagnostic and follow-up of HTN, compared with office BP, despite huge amount of evidence in favor of ABMP or HBMP. A whole article should be dedicated as to which group of professionals is the most adequate to write professional guidelines. 

We summarized these considerations and added two short sentence page 6, line " Finally, a major source of inconsistencies within, and discrepancies between guidelines can be found in conflicts of interest affecting contributors (REF), COI can be of different nature, industrial (often disclosed), economic, academic and institutional (almost never disclosed), but in any case tend to bias the guidelines. Although imperfect, the present methodology is next to best (REF)"

We apologize for the broken English. We had the paper reviewed by a native English speaking researcher, and hope to have raised the paper to the expected standards.

Reviewer 2 Report

The authors have written a review article on hypertension control guidelines by three pre eminent societies: AHA/ACC, ESC and Canadian guidelines. Difference in diagnosis of arterial hypertension and treatment strategy amongst the societies are discussed.

Overall, it is an interesting read, I have few comments.

  1. I recommend the authors choose a narrative and objective direction of the review article. in page 3, line 84-85 the inability of guidelines to provide a consensus statement on the best methodology of blood pressure measurement is discussed. Please provide diagnostic data with certain gold standards, to demonstrate difference in BP measurement in standards and controls.
  2. Management options are discussed, while delving into the sprint trial, however the authors have not elaborated the impact of diabetes, and ASCVD which have been discussed in a different context by different societies. Please provide a tabulated version of difference in management approaches amongst the three guidelines.
  3. Please include the 2020 International Society of Hypertension guidelines as a comparator also to cover a wider audience.
  4. The authors have delved into usage of PVW and markers of end organ damange as surrogates for BP. Request improved discussion into sensitivity, specificity, relevant cutoffs for the markers and outcomes. A tabulation would make this comprehensive.

Author Response

We thank the reviewer for his(her) positive remarks. 

Point 1: we have rephrased the objectives. In the scope of the present paper, it was not our intention to make an exhaustive review, point by point on similarities and differences. This has been done by others before. Our objective was mainly to stress that the differences between guidelines are more a consequence of competition between learned societies, than based on scientific evidences. Second, that all guidelines are biased by economic and academic considerations, which explain why all 4 guidelines are very conservative and neglect major scientific advances (such as out of office BP measures or integration of HMOD in clinical care). We tried to put that in a diplomatic way.  

Point 2 : We have included diabetes in the paper. by adding a short paragraph and 3 references 

POint 3: We have included ISH guidelines. The first version did not because those guidelines are really overlapping much with ESH, many contributors being the same

Point 4: PWV was given as an example of what the guidelines are missing. We could have done the same with LVH, microalbuminuria, IMT etc. We have edited the paragraph to make it blend more in the rest of the paper, and better articulation in the narrative. 

Reviewer 3 Report

Authors described the resemblance, divergences and incosistencies of international guidelines for hypertension. I have nothing to comment. The manuscript was well written. I have nothing to comment.

Author Response

We thank the reviewer for her(his) positive opinion on the paper